# Validation Study of Italian Version of Inventory for Déjà Vu Experiences Assessment (*I-IDEA*): A Screening Tool to Detect Déjà Vu Phenomenon in Italian Healthy Individuals

**DOI:** 10.3390/bs7030050

**Published:** 2017-08-06

**Authors:** Laura Mumoli, Giovanni Tripepi, Umberto Aguglia, Antonio Augimeri, Rossella Baggetta, Francesca Bisulli, Antonella Bruni, Salvatore M. Cavalli, Alfredo D’Aniello, Ornella Daniele, Carlo Di Bonaventura, Giancarlo Di Gennaro, Jinane Fattouch, Edoardo Ferlazzo, Alessandra Ferrari, Annateresa Giallonardo, Sara Gasparini, Salvatore Nigro, Andrea Romigi, Vito Sofia, Paolo Tinuper, Maria Grazia Vaccaro, Leila Zummo, Aldo Quattrone, Antonio Gambardella, Angelo Labate

**Affiliations:** 1Institute of Neurology, University Magna Græcia, 88100 Catanzaro, Italy; laura.mumoli@gmail.com (L.M.); u.aguglia@unicz.it (U.A.); bruni_antonella@yahoo.it (A.B.); jurijz@hotmail.it (S.M.C.); edoferl@hotmail.it (E.F.); s.gasparini@neurorc.it (S.G.); vaccaro.mg@gmail.com (M.G.V.); a.quattrone@unicz.it (A.Q.); a.gambardella@unicz.it (A.G.); 2Institute of Clinical Physiology, National Research Council (IFC-CNR), Research Unit, 89100 Reggio Calabria, Italy; gtripepi@hotmail.com (G.T.); rossella.baggetta@gmail.com (R.B.); 3Regional Epilepsy Center, Bianchi Melacrino Morelli Hospital, 89100 Reggio Calabria, Italy; 4Institute of Molecular Bioimaging and Physiology of the National Research Council (IBFM-CNR), 88100 Catanzaro, Italy; antonio.augimeri@bioteconmed.it (A.A.); salvatoreangelo.nigro@gmail.com (S.N.); 5IRCCS Istituto delle Scienze Neurologiche, 40100 Bologna; Department of Biomedical and Neuromotor Sciences, University of Bologna, 40126 Bologna, Italy; francesca.bisulli@unibo.it (F.B.); paolo.tinuper@unibo.it (P.T.); 6IRCSS Neuromed, 86077 Pozzilli, Italy; alfredod@vodafone.it (A.D.); gdigennaro@neuromed.it (G.D.G.); 7Experimental Biomedicine and Clinical Neuroscience Department (BioNeC), University of Palermo, 90100 Palermo, Italy; ornella.daniele@unipa.it (O.D.); leilazummo@yahoo.it (L.Z.); 8Department of Neuroscience, Neurology Unit, “Sapienza” University, 00100 Rome, Italy; c_dibonaventura@yahoo.it (C.D.B.); jinafatti@yahoo.it (J.F.); annaterasa.giallonardo@uniroma1.it (A.G.); 9Clinical Neurophysiology, Department of Neuroscience, Ophthalmology and Genetics, University of Genoa, 16100 Genova, Italy; ferrarialessandrarousseau@gmail.com; 10Sleep Medicine Centre, Neurophysiopathology Unit, Department of Systems Medicine, “Tor Vergata” University of Rome, 00100 Rome, Italy; a_romigi@inwind.it; 11Department “G. F. Ingrassia” University of Catania, Via S. Sofia 78, 95123 Catania, Italy; vitosofia55@gmail.com

**Keywords:** Déjà vu, Inventory for Déjà Vu Experiences Assessment (IDEA), prevalence of DV

## Abstract

The Inventory Déjà Vu Experiences Assessment (IDEA) is the only screening instrument proposed to evaluate the Déjà vu (DV) experience. Here, we intended to validate the Italian version of IDEA (I-IDEA) and at the same time to investigate the incidence and subjective qualities of the DV phenomenon in healthy Italian adult individuals on basis of an Italian multicentre observational study. In this study, we report normative data on the I-IDEA, collected on a sample of 542 Italian healthy subjects aging between 18–70 years (average age: 40) with a formal educational from 1–19 years. From September 2013 to March 2016, we recruited 542 healthy volunteers from 10 outpatient neurological clinics in Italy. All participants (i.e., family members of neurological patients enrolled, medical students, physicians) had no neurological or psychiatric illness and gave their informed consent to participate in the study. All subjects enrolled self-administered the questionnaire and they were able to complete I-IDEA test without any support. In total, 396 (73%) of the 542 healthy controls experienced the DV phenomenon. The frequency of DV was inversely related to age as well as to derealisation, jamais vu, precognitive dreams, depersonalization, paranormal activity, remembering dreams, travel frequency, and daydreams (all *p*
< 0.012). The Italian version of IDEA maintains good properties, thus confirming that this instrument is reliable for detecting and characterising the DV phenomenon.

## 1. Introduction

The Déjà vu (DV) experience is defined as “any subjectively erroneous feeling of familiarity for present experience with an undefined past” [1]. Recently, the DV phenomenon has aroused significant interest in scientific literature because many aspects of DV are still obscure. In the current literature, there is no unique definition universally accepted to describe DV, and its significance remains unclear, even if the physiopathology of DV is known, as it has been investigated mainly in patients with temporal lobe epilepsy (TLE) [2,3].

Although DV was firstly described back in 1896 [4], there are no standardised instruments to screen DV in each country, as in Italy. To date, a scale-self-administered questionnaire called the Inventory Déjà Vu Experiences Assessment (IDEA) is the only instrument validated and able to measure the frequency of occurrence and the psycho-behavioural consequences of having DV [5]. 

The IDEA represents a valuable method to explore the impact and the features of DV. It is a 23-item self-administered questionnaire containing a general section of nine questions and a qualitative section of 14 sections focusing on the qualitative characteristic of the DV experiences. The IDEA test has been translated and validated in Japan [6]. In this paper, our purpose is to validate the Italian version of the IDEA (I-IDEA) and at the same time investigate the incidence of the DV phenomenon among healthy Italian adult individuals.

For this reason, the IDEA test was translated from English into Italian and approved by the original author, Sno. Here, we report the validation of the I-IDEA test as well as the preliminary data analysis on the prevalence of DV in an ongoing Italian multicentre observational study/survey. The possibility to have a validated instrument to collect DV information offers a unique chance to study both normal and pathological DV.

## 2. Methods

The study was approved by the Ethic Committee of University of Catanzaro. All data collected from the other 10 outpatient neurological clinics in Italy (Epilepsy Centre Hospital Bianchi-Melacrino Reggio Calabria, “Tor Vergata” University of Rome, “Sapienza” University, Rome, IRCSS Neuromed, Pozzilli, University of Palermo, University of Catania, IRCCS and University of Bologna, University of Genoa) were sent to our Unit (University of Catanzaro) and included in a common and homogeneous database. In accordance with the international guidelines for translation and cross-cultural adaptation [7], the original version of IDEA was translated into Italian by L.M.; then a native English–speaking reviewer translated the Italian version in back into English. This latter form was compared with original one, and subsequently the original author, Sno, approved the translated version of the Italian I-IDEA. The translation procedure was carried out according to accepted international standards [8,9]. The original English version was forward-translated by two independent translators—an English native speaker teacher and a doctor fluent in English—and their translation agreed with a final Italian version. This first Italian version was independently back-translated into English by another translator and by a psychologist fluent in English with experience in research; these versions, in turn, agreed on a final English back-translation. The Italian translation and the English back-translation were then reviewed by a multi-disciplinary committee composed of a professor of Neurology, a psychologist with experience in a research unit, a physician, and a psychologist. The English back-translation was compared to the original version in order to detect any misinterpretation and ambiguity; the two versions were found to be reasonably similar. Furthermore, the Italian translation was compared to the original version to ensure conceptual equivalence and improve understandability.

As showed in Figure 1, this is a multicenter, cross-sectional study involving ten hospitals widespread in the Italian territory. The participants were all native Italian speakers and were mainly family members of patients coming to the neurology clinic, medical students, and staff from each hospital. Exclusion criteria were people younger than 18 years and people with previous or current medical history including neurological (especially epilepsy) or psychiatric illness. All participants were given detailed oral and written informed consent to participate in the study. All subjects enrolled self-administered and completed the IDEA test. In this study, the psychological experiences in part A of the IDEA items (i.e., DV, derealisation, paranormal quality, remembering dreams, travel frequency, daydreams) were analysed mainly because all participants responded to these questions. According to the IDEA, if subjects checked “Don’t know,” it was regarded as “never” and they did not continue to part B. With respect the original IDEA, we also added information about handedness analysed by the Annett Hand Preference test [10] to evaluate the potential lateralized effects on the development of DV experiences. We also collected information about the rate of age and education.

### Statistical Analysis

Data are summarised as mean and standard deviation or as percent frequency, as appropriate. The Cronbach’s α value used as a criterion of adequate internal consistency reliability was 0.70 or higher. Data analysis was performed by SPSS for Windows (version 22.0, IBM, Chicago, IL, USA). 

The relationships between age and the IDEA items were investigated by Spearman rank correlation coefficients (rho) and P values. To assess the weight of Déjà vu among the various mental phenomena, a factorial analysis (Varimax rotation) was used with a principal component solution. Initial un-rotated factors were obtained by principal component methods, and those with an eigenvalue >1 underwent Varimax rotation.

## 3. Results

In total, 542 native (232 men and 310 women; age 40 ± 20 years) Italian–speaking healthy controls were collected and examined by trained neurologists using the I-IDEA test. The internal consistency reliability of the questionnaire was satisfactory because the Cronbach’s α value was 0.7. The full list of questions and the corresponding items of the I-IDEA test are detailed in Table 1. On the basis of the Annett Hand Preference Questionnaire, 95% of our population were right handers, 4.5% were left-handers, and 0.5% were mixed.

### 3.1. Frequency of Déjà Vu and Related Experiences

According to the first question of the I-IDEA test, 73% of subjects stated that they experienced “recognition” (the fact of knowing someone or something already) during their life. Among these, as many as 7.7% revealed that the frequency of this phenomenon ranged from few times a month to at least weekly. In questions 2 and 3, a substantial proportion of subjects answered that, from few times a year to at least weekly, they feel that it seems as though everything around is not real (17.9%) or that they had never experienced something before when in fact they had experienced it before (13.5%). In question 4, 28% of individuals revealed that they really experienced something that had occurred before in a dream (with a frequency ranging from sometime to very frequently), and such an answer was accompanied by the statement that, while something was happening to them, they felt that it was not happening to themselves but to someone else, in a substantial proportion of subjects with a frequency ranging from sometime to more frequently (11.6%). Remarkably, in question 6, 3.9% of subjects stated that they consider their self as persons with paranormal qualities. The answers to the remaining questions (questions 7 and 8) are given in Table 1. 

The second part of Table 2 reports further answers from subjects who provided “Yes” to the first question of the questionnaire (i.e., subjects who experienced a recognition with a frequency ranging from very infrequently to more frequently (n = 396 individuals)]. The most frequent answer to the first question (I-IDEA test—part 2) was that they experienced a recognition “in a certain situation” (73.9%) followed by “in a certain place” (70.4%), “at a certain event” (65.6%), “when meeting someone” (57.5%), while “engaging a certain activity” or “when telling someone about something” (53.9%), and less frequently “while listening to a conversation, music, or a statement” (45.8%), while “having a certain thought” (41.3%), “while reading something” (30.4%), or “in some other way” (3.8%). Of note, 12.2% of those interviewed exactly remember where and when they had a recognition and this answer was in keeping with the next one (see question 3), in which as much as 19.5% of subjects stated that they remember the recognition occurred 1–5 years ago. The large majority of subjects (66.1%) answered that the duration of recognition was a few seconds, whereas as many as 19.5% stated that the duration of the same phenomenon was from one to a couple of minutes (see question 4). The recognition usually related to some part of an experience or situation in 57% of answers (see question 5), but 71.6% of subjects stated that they did not habitually experience recognition at a certain time of day (see question 5). In question 7, as much as 16.5% of subjects stated that while having a recognition, they feel that they can predict what is going to happen in the next few minutes, with a frequency ranging from sometimes to more frequently. In question 8, the majority of subjects (68.8%) who experienced recognition stated that they do not have the feeling that it was not happening to them but to someone else, and 22.5% of subjects (see question 9) stated that they have the feeling that the recognition usually pertains to an exact repetition of the past. In question 10, a substantial proportion of subjects (40.5%) stated that the recognition was accompanied by the feeling as if everything around was not real, and as many as 57% of the subjects stated that the recognition is a surprising and amazing experience (see question 11). Of note, 10.9% of those interviewed stated that reincarnation and paranormal qualities (see question 12) are the most acceptable explanations of the recognition that in the majority of circumstances (49.9%) occurred while the subject was relaxed (see question 13) or during a concentrated activity (42%—see question 14).

### 3.2. Relationships between Age and IDEA Items

As shown in Table 2, in our study, age was strongly and inversely related to Déjà vu: Déjà vu decreases as age increases. Age was also inversely related to remembering dreams, travel frequency, precognitive dreams, and daydreaming. Age tended to also be related to Jamais vu, but it was largely unrelated to derealisation, depersonalization, and paranormal activity (Table 2). A face-to-face comparison of the effect of age on each IDEA item between ours (r ranging from −0.380 to 0.005) and Adachi’s study (r ranging from −0.318 to 0.066) showed that the Déjà vu–age link was of similar strength between the two studies. In both reports, age was also related to precognitive and remembered dreams. 

### 3.3. Relationships among IDEA Items

Interrelationships among IDEA items are given in Table 3. As reported in Table 3, precognitive dreams were most correlated of Déjà vu (rho = 0.296, *p* < 0.001), followed by derealization (rho = 0.248, *p* < 0.0001) and daydreams (rho = 0.24, *p* < 0.001). The remaining intercorrelations among the various IDEA items ranged from 0.005 (Derealization versus Remembering dreams) to 0.344 (Derealization versus Depersonalization) (*p* values ranging from <0.001 to 0.915). The strength of interrelationships among the IDEA items in our study were all of a similar degree to those observed in Adachi’s study. 

### 3.4. Factorial Analysis of IDEA Items

According to the previous paper by Adachi et al. [6], the nine items of the IDEA test were reduced by factorial analysis into three experience profiles: dissociated-related items, dream-memory-related items, and mental activity-related items (Table 4). Derealization (rho = 0.660, *p* < 0.001) was the most important item associated to Factor 1 (Dissociation-related item), followed by Daydreams (rho = 0.652, *p* < 0.001), Depersonalization (rho = 0.617, *p* < 0.001), and Jamais vu (rho = 0.563, *p* < 0.001). Factor 2 (Dream/memory-related item) resulted to be closely related to Déjà vu (rho = 0.833, *p* < 0.001) and Precognitive dreams (rho = 0.727, *p* < 0.001), and only slightly associated to Remembering dreams (rho = 0.207, *p* < 0.001). Factor 3 (Mental activity-related item) was linked to Travel frequency (rho = 0.778, *p* < 0.001) and Paranormal quality (rho = 0.591, *p* < 0.001). The interrelationships between variables considered in the factorial analysis in our study were generally of smaller degree as compared to those found by Adachi [6].

## 4. Discussion

The I-IDEA is the second officially approved non–English translation of the IDEA (pdf version available to download, see Appendix A). In 2001, Adachi et al. [6] published the Japanese version of the IDEA for which validity and reliability were equivalent to those of the English version. The purpose of the present work was to give physicians a valid and easily accessible instrument to screen DV in an out-patient ambulatory of neurology.

In this study, we also depicted the demographic and psychological features of DV in an Italian population. The I-IDEA test presented here reaches the criteria to be defined as an official translation of the IDEA, and our wish is to propose that this test be freely accessible for clinical purposes mainly in order to give users the opportunity to quickly identify DV experience according to a common, shared definition and to avoid potential misdiagnosis.

The results of this study clearly show that the I-IDEA has excellent internal consistency and reproducibility, which are comparable to the original scale. 

The vast majority (73%) of our healthy population had DV experiences, as previous reported by several studies in healthy subjects, with a frequency of DV experience ranging from 31 to 96% [6,7,11,12].

Considering that the DV phenomenon is very often present in patients with mesial temporal lobe epilepsy, this paper will be a practicable instrument to investigate the DV phenomenon in patients with epilepsy and clarify whether DV is different between healthy subjects and individuals with epilepsy. For this reason, this paper is propaedeutic to future studies in this research field. Obviously, there are some limitations that must be considered: First, potential cognitive or neuropsychological issues cannot fully be excluded because people recruited were not screened with appropriate behavioural assessment. However, each individual was prior interviewed by a trained neurologist (e.g., consultant or professor of neurology) with particular experience in epilepsy. Second, our findings need to be confirmed in other studies.

Moreover, the I-IDEA is a reliable and user-friendly instrument that is able to measure qualitative and quantitative DV phenomena, and it could be used particularly in epilepsy or psychiatric centres to objectively screen DV. 

In close parallelism with the results of Adachi et al. [6], age was strongly and inversely related to Déjà vu. Similarly, the interrelationships among IDEA items found in our study were very similar to those that emerged in the Adachi’s study [6], indicating that the Italian translation of the IDEA test has a satisfactory internal and external consistency.

## Figures and Tables

**Figure 1 behavsci-07-00050-f001:**
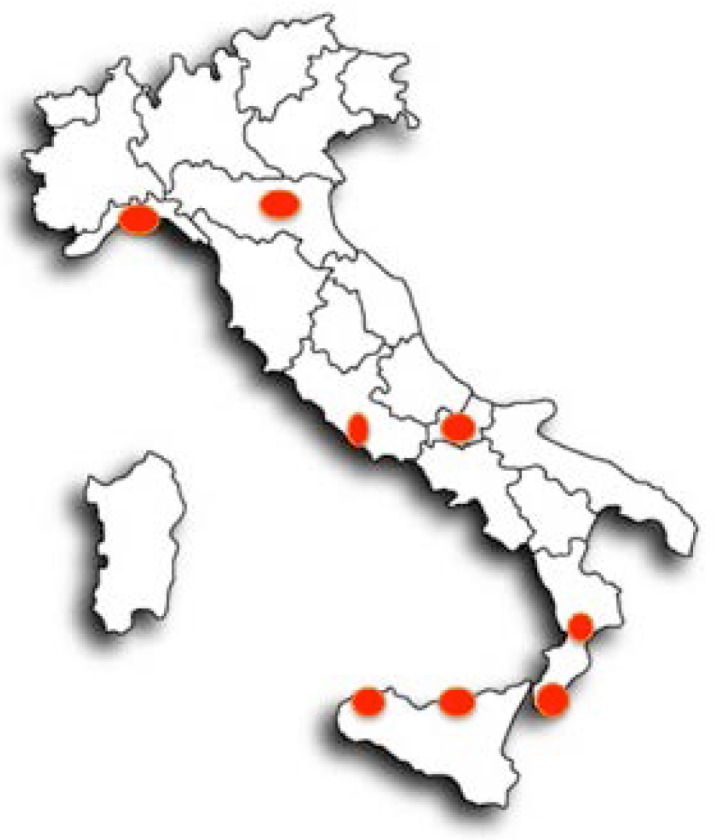
Italian centers participating to the study.

**Table 1 behavsci-07-00050-t001:** Percentage of answers of each Italian Inventory Déjà Vu Experiences Assessment (I-IDEA) question.

I-IDEA TEST 1st Part (N = 542 CONTROLS)	Item	(%)
1 Have you ever had the feeling of having experienced a sensation or situation before in exactly the same way when in fact you are experiencing it for the first time?	O Never	25.3
	O Yes, very infrequently (less than once per year)	22.5
	O Yes, sometimes (a few times per year)	42.8
	O Yes, often (a few times a month)	6.8
	O Yes, more frequently (at least weekly)	0.9
	O Don’t know	1.7
2 Have you ever had the feeling that it seems as if everything around is not real, as if it is not really happening?	O Never	60.0
	O Very infrequently (less than once per year)	19.4
	O Sometimes (a few times a year)	15.7
	O Often (a few times a month)	1.8
	O More frequently (at least weekly)	0.4
	O Don’t know	2.8
3 Note: This question is about the opposite of the feeling of “recognition.” Have you ever had the feeling that you had never experienced something before, when in fact you had experienced it before? For example: You see something or someone you know very well, but you feel as if you have never seen it or him before!	O Never	69.4
	O Very infrequently (less than once per year)	12
	O Sometimes (a few times a year)	11.8
	O Often (a few times a month)	1.3
	O More frequently (at least weekly)	0.4
	O Don’t know	5.2
4 Has it ever happened to you that you experienced something that had occurred before in a dream?	O Never	34.3
	O Very infrequently (less than once per year)	33.2
	O Sometimes (a few times a year)	22.1
	O Often (a few times a month)	5.7
	O More frequently (at least weekly)	0.2
	O Don’t know	4.4
5 Have you ever had the feeling while something was happening to you that it was not happening to yourself, but to someone else, as if you were looking at yourself?	O Never	67.5
	O Very infrequently (less than once per year)	18.8
	O Sometimes (a few times a year)	10.1
	O Often (a few times a month)	1.3
	O More frequently (at least weekly)	0.2
	O Don’t know	2
6 Do you consider yourself a person with paranormal qualities? *(‘Paranormal qualities’ includes clairvoyance, telepathic or psychic abilities and so forth.)*	O No	81.9
	O No, but I am not sure	6.6
	O Yes, but I am not sure	5.9
	O Yes	3.9
	O Don’t know	1.7
7 How often can you remember a dream so well that you can tell someone about it?	O Never	6.5
	O Very infrequently (less than once per year)	16.4
	O Sometimes (a few times a year)	33.6
	O Often (a few times a month)	23.8
	O More frequently (at least weekly)	17.9
	O Don’t know	1.8
8 How many times a year do you travel a distance of about a hundred kilometres or more from your home locality?	O Never	4.8
	O Very infrequently (less than once per year)	15.3
	O Sometimes (a few times a year)	35.4
	O Often (a few times a month)	22.9
	O More frequently (at least weekly)	20.8
	O Don’t know	0
9 Has it ever happened to you that you were daydreaming?	O Never	45.9
	O Very infrequently (less than once per year)	17.5
	O Sometimes (a few times a year)	19.2
	O Often (a few times a month)	8.9
	O More frequently (at least weekly)	5
	O Don’t know	3.3
**I-IDEA TEST 2nd Part (These Data Refer Only to Individuals Who Answered “Yes” to the First Question, i.e., in 396 Individuals)**		**(%)**
1 A person can have a feeling of “recognition” in many different ways. It can have to do with a specific place, a situation, an activity, an event, meeting someone, a conversation, a thought, reading a book or a newspaper, Have you ever had this feeling of “recognition” in one or more of the following ways? (Note: You can answer ‘Yes’ to more than one topic of this question, Please answer all the topics, including the ones you answer ”No.” If you are not sure whether something is applicable to you, answer “No.”)	a, In a certain **place**	70.4
	b, In a certain **situation**	73.9
	c, Engaging in a certain **activity**	53.9
	d, At a certain **event**	65.6
	e, When **meeting** someone	57.5
	f, While **telling** someone about something	53.9
	g, While **listening** to a conversation, music, or a statement	45.8
	h, While having a certain **thought**	41.3
	i, While **reading** something	30.4
	j, In some **other way** than in question a-i,	3.8
2 While you have this feeling of “recognition,” can you remember exactly where and when you had the same experience or feeling before?	O No	41.5
	O I vaguely remember	40.8
	O Yes, I can remember exactly	12.2
	O Don’t know	5.6
3 When did this feeling of “recognition” occur for the last time?	O More than 5 years ago	9.6
	O 1 to 5 years ago	19.5
	O 6 months to 1 year ago	26.6
	O 2 to 6 months ago	11.4
	O 1 to 2 months ago	9.6
	O Last month	9.6
	O Don’t know	13.7
4 How long does this feeling of “recognition” usually last?	O One second or less	8.9
	O A few seconds	66.1
	O One minute or a couple of minutes	19.5
	O Half an hour to one hour	0.8
	O A few hours	0.3
	O More than a few hours	0.3
	O Don’t know	4.3
5 Is the feeling of “recognition” usually related to some part of an experience or situation, or to the whole thing?	O Total	11.9
	O Some part of it	57.0
	O It depends	15.9
	O Don’t know	15.2
6 Do you usually have this feeling of “recognition” at a certain time of day?	O No	71.6
	O In the morning shortly after awakening	2.5
	O In the Daytime	7.8
	O When it gets dark	2.0
	O In the evening (with the lights on)	0.5
	O Just before or after going to bed	1.3
	O Don’t know	14.2
7 While having this feeling of *‘recognition’,* did you ever have the idea you could predict what was going to happen in the next few minutes?	O Never	58.5
	O Very infrequently (less than once per year)	16.7
	O Sometimes (a few times a year)	11.9
	O Often (a few times a month)	3.3
	O More frequently (at least weekly)	1.3
	O Don’t know	8.4
8 While having this feeling of “recognition,” did you ever have the feeling it was not happening to you but to someone else, as if you were looking at yourself?	O No	68.8
	O Vague feeling it was not happening to me	9.1
	O Clear feeling it was not happening to me	0.5
	O Vague feeling I was looking at myself	9.9
	O Clear feeling I was looking at myself	3.6
	O Don’t know	8.1
9 Does this feeling of “recognition” usually pertain to an exact repetition of the past or to approximately the same thing?	O Exactly the same	12.4
	O Almost exactly the same	22.5
	O The same	6.8
	O Approximately the same	30.1
	O Vaguely the same	13.2
	O Don’t know	14.9
10 While having this feeling of “recognition” have you also ever felt that it looked as if everything around you was not real, as if it was not really happening?	O Never	50.9
	O Yes, a little unreal	26.1
	O Yes, vaguely unreal	10.1
	O Yes, unreal	3.5
	O Yes, totally unreal	0.8
	O Don’t know	8.6
11 In general, how does this feeling of “recognition” affect you?	a, It leaves me indifferent	45.1
	b, It frightens me	13.2
	c, It is reassuring	13.7
	d, It is nice and pleasant	29.9
	e, It is uncomfortable or oppressive	9.4
	f, It is surprising, amazing	57.0
	g, It interrupts whatever I am doing	25.1
	h, Other effect:	2.3
12 What do you feel is the explanation of this feeling of “recognition”?	a, Anxiety or tension	21.5
	b, Poor memory	21.3
	c, Unconscious memories	65.3
	d, Reincarnation	5.8
	e, Concentration problems	22.5
	f, Paranormal qualities	5.1
	g, Desire to escape from reality	19.5
	h, Other explanation:	5.3
13 How do you usually feel before you have this feeling of “recognition”?	a, Mentally fatigued	13.9
	b, Gloomy or depressed	9.9
	c, Nervous or under stress	13.2
	d, Physically fatigued	12.9
	e, Cheerful and happy	16.5
	f, Confused or absent-minded	14.2
	g, Relaxed	49.9
	h, Angry	4.8
	i, Frightened	5.8
	j, Drowsy	10.4
	k, Physically ill	2.0
14 Have you ever had this feeling of “recognition” in one of the following conditions?	a, Headache	7.1
	b, ‘Black out’	9.6
	c, Epileptic seizure	0.3
	d, Concentrated activity	42.0
	e, Drinking alcohol	6.3

**Table 2 behavsci-07-00050-t002:** Correlation between age and the IDEA items.

Experiences	Age	
	Spearman’s Coefficient (rho) and *p* Value	
	Adachi’s study [6]	Present study
Deja vu	**−0.380 (<0.001)**	**−0.318 (<0.001)**
Derealization	**−0.165 (0.001)**	−0.049 (0.252)
Jamais vu	0.005(0.919)	−0.084 (0.051)
Precognitive dreams	**−0.244 (<0.001)**	**−0.106 (0.014)**
Depersonalization	−0.087 (0.088)	−0.040 (0.355)
Paranormal activity	−0.005 (0.915)	0.066 (0.124)
Remembering dreams	**−0.173 (0.001)**	**−0.241 (<0.001)**
Travel frequency	−0.043 (0.399)	**−0.173 (<0.001)**
Daydreams	−0.009 (0.858)	**−0.100 (0.020)**

Statistically significant associations are in bold.

**Table 3 behavsci-07-00050-t003:** Relation between the IDEA items.

Experiences	Deja Vu	Derealization	Jamais Vu	Precognitive Dreams	Depersonalization	Paranormal Activity	Remembering Dreams	Travel Frequency
**Derealization**								
*rho*	0.248 (0.301)							
*p*	<0.001							
**Jamais vu**								
*rho*	0.169 (0.128)	0.216 (0.330)						
*p*	<0.001	<0.001						
**Precognitive dreams**								
*rho*	0.296 (0.376)	0.288 (0.276)	0.164 (0.102)					
*p*	<0.001	<0.001	<0.001					
**Depersonalization**								
*rho*	0.188 (0.128)	0.344 (0.436)	0.208 (0.329)	0.211 (0.204)				
*p*	<0.001	<0.001	<0.001	<0.001				
**Paranormal activity**								
*rho*	0.107 (0.087)	0.193 (0.132)	0.10 (0.095)	0.143 (0.158)	0.141 (0.195)			
*p*	0.012	<0.001	0.02	0.001	0.001			
**Remembering dreams**								
*rho*	0.183 (0.246)	0.005 (0.134)	−0.026 (0.090)	0.182 (0.239)	0.031 (0.179)	0.137 (0.170)		
*p*	<0.001	0.915	0.544	<0.001	0.478	0.001		
**Travel frequency**								
*rho*	0.109 (0.095)	0.002 (0.115)	0.026 (0.109)	0.018 (0.012)	−0.008 (0.125)	0.016 (0.096)	0.158 (0.068)	
*p*	0.011	0.969	0.553	0.671	0.86	0.717	<0.001	
**Daydreams**								
*rho*	0.24 (0.152)	0.243 (0.202)	0.201 (0.209)	0.189 (0.200)	0.235 (0.144)	0.195 (0.131)	0.077 (0.092)	0.087 (0.06)
*p*	<0.001	<0.001	<0.001	<0.001	<0.001	<0.001	0.074	0.043

Data are Sperman rank correlation (rho) and *p* values. In parenthesis, the rho coefficient between each pair of items as reported in the Adachi’s study [6] is also given.

**Table 4 behavsci-07-00050-t004:** Factor analysis of the IDEA Items.

Items	Factor 1	Factor 2	Factor 3
**Dissociation-related**			
Derealization	Rho = 0.660 (0.749)		
	*p* < 0.001		
Jamais vu	rho = 0.563 (0.749)		
	*p* < 0.001		
Depersonalization	rho = 0.617 (0.677)		
	*p* < 0.001		
Daydreams	rho = 0.652 (0.534)		
	*p* < 0.001		
**Dream/memory-related**			
Precognitive dreams		rho = 0.727 (0.740)	
		*p* < 0.001	
Deja vu		rho = 0.833 (0.714)	
		*p* < 0.001	
Remembering dreams		rho = 0.207 (0.673)	
		*p* < 0.001	
**Mental activity-related**			
Travel frequency			rho = 0.778 (0.772)
			*p* < 0.001
Paranormal quality			rho = 0.591 (0.616)
			*p* < 0.001

Data are Sperman rank correlation (rho) and *p* values. In parenthesis, the rho coefficient between each pair of items as reported in the Adachi’s study [6] is also given.

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
