# Peer review of "Validation Study of Italian Version of Inventory for Déjà Vu Experiences Assessment (I-IDEA): A Screening Tool to Detect Déjà Vu Phenomenon in Italian Healthy Individuals"

_behavsci, 2017, doi:10.3390/bs7030050_

Round 1

Reviewer 1 Report

Authors overcome all my criticism and followed my suggestions.

Author Response

Thank you for your comments

Reviewer 2 Report

It is clear that the authors have gone to substantial effort to address the concerns raised in the previous round of reviews, which is appreciated. However there are still some issues in terms of content and presentation, as per below:

line 62 - The broader relevance of this point needs explaining and contextualising much more clearly as at present it isn't really that clear as to why it needs mentioning.

line 135 - 'experienced a recognition' and subsequent references to 'recognition' - this has the potential to confuse the reader as there are now no explicit references to deja vu.

line 178 - wouldn't it make more sense to say that deja vu decreases as age increases?

line 182 - you say there is a strong link between this and the Adachi study - it would be worth reporting the size of their correlation to enable direct comparison.

line 217 - not really sure what is meant by a 'unique definition' of deja vu - surely the IDEA should reflect the use of a common definition?

line 233 - the comment re: sample size is quite vague - 'rather large' isn't particularly objective/scientific.

Whilst it is clear that some effort has gone into improving the English in this manuscript, there are still issues of clarity of expression that need to be addressed throughout and so I feel it necessary to highlight the need for additional proof reading.

Author Response

Response to Reviewer 2

line 62 - The broader relevance of this point needs explaining and contextualising much more clearly as at present it isn't really that clear as to why it needs mentioning.

We delete the phrase that was confounding as the reviewer observed.

line 135 - 'experienced a recognition' and subsequent references to 'recognition' - this has the potential to confuse the reader as there are now no explicit references to deja vu.

We better explained the meaning of the word recognition in brackets.

line 178 - wouldn't it make more sense to say that deja vu decreases as age increases?

We corrected the phrase accordingly.

line 182 - you say there is a strong link between this and the Adachi study - it would be worth reporting the size of their correlation to enable direct comparison.

The range of correlation coefficients is now reported in ours and in the Adachi’s study.

line 217 - not really sure what is meant by a 'unique definition' of deja vu - surely the IDEA should reflect the use of a common definition?

We changed the word and used the term common shared as the reviewer observed.

line 233 - the comment re: sample size is quite vague - 'rather large' isn't particularly objective/scientific.

The referee is right. We removed part of the sentence.

Whilst it is clear that some effort has gone into improving the English in this manuscript, there are still issues of clarity of expression that need to be addressed throughout and so I feel it necessary to highlight the need for additional proof reading.

A mother-language reviewed the English language.